# Functioning and Happiness in People with Schizophrenia: Analyzing the Role of Cognitive Impairment

**DOI:** 10.3390/ijerph18147706

**Published:** 2021-07-20

**Authors:** Luis Gutiérrez-Rojas, Pablo Jose González-Domenech, Gema Junquera, Tate F. Halverson, Guillermo Lahera

**Affiliations:** 1Department of Psychiatry, Faculty of Medicine, University of Granada, 18016 Granada, Spain; pgdomenech@gmail.com; 2Department of Medicine and Medical Specialities, Faculty of Medicine and Health Sciences, University of Alcalá, 28801 Alcalá de Henares, Spain; gemajunqueraf@gmail.com (G.J.); guillermo.lahera@gmail.com (G.L.); 3Mid-Atlantic Mental Illness Research, Education & Clinical Center, Durham Veterans Affairs Health Care System, Durham, NC 27705, USA; tate.halverson@unc.edu; 4Ramon y Cajal Institute of Sanitary Research (IRYCIS), 28034 Madrid, Spain; 5Psychiatry Service, Center for Biomedical Research in the Mental Health Network, University Hospital Principe de Asturias, 28806 Alcala de Henares, Spain

**Keywords:** schizophrenia, happiness, functioning, cognitive impairment, perceived stress, satisfaction with life

## Abstract

Schizophrenia is associated with marked functional impairment and low levels of subjective happiness. The aim of the current study was to evaluate the relationship between subjective happiness and functioning in patients with schizophrenia, while considering the role of cognitive functioning. Methods: In total, 69 schizophrenia patients and 87 matched healthy controls participated in the study. Patients’ clinical status was assessed, and a series of self-report questionnaires were administered to both patients and healthy controls to measure subjective happiness, satisfaction with life, well-being, functioning, and cognitive impairment. A multiple linear regression model identified significant predictors of subjective happiness and related constructs. Results: Schizophrenia participants endorsed lower levels of happiness and well-being, and higher perceived stress compared to healthy controls. In schizophrenia patients, there was an inverse and significant correlation (r = −0.435; *p* = 0.013) between subjective happiness and functioning in a subgroup of patients without cognitive impairment. This correlation was not significant (r = −0.175; *p* = 0.300) in the subgroup with cognitive impairment. When controlling for other clinical variables (by multiple lineal regression), the severity of symptoms and level of insight failed to demonstrate significant relationships with happiness; meanwhile, perceived stress and some specific cognitive dominions (as verbal learning and processing speed) were associated with satisfaction of life of the patients. Conclusions: The relationship between subjective happiness and functioning in schizophrenia patients was influenced by level of cognitive impairment. Findings from this study suggest that rehabilitation programs may improve recovery outcomes with a focus on subjective happiness and functioning, especially in patients with cognitive impairment. Future research is needed to better understand the complex interplay between subjective happiness, functioning, and cognitive impairment in patients with schizophrenia.

## 1. Introduction

Schizophrenia is a chronic, severe, and often stigmatizing mental illness characterized by the presence of cognitive impairments and deficits in motivation that adversely impact overall functioning [1,2,3].

The negative impact of schizophrenia on quality of life [4] makes this disease one of the main causes of disability worldwide [5]. Unlike other causes of disability, schizophrenia has the added disadvantage of social stigma [1]; for example, the general population may perceive individuals with schizophrenia as dangerous and violent, who cannot maintain consistent employment to support themselves, are not reliable, and may not be able have children. The stigma experienced by patients is a complex construct related to numerous factors such as: the presence of cognitive impairment [6], psychosocial disability [7], lack of social and family support, and significant alterations in functioning [8].

Bradburn (1969) [9] and Argyle (1987) [10] defined happiness as the global balance of positive and negative affects throughout the lifespan where the positive affect exceeds the negative affect. To date, there are few studies specifically focused on understanding happiness in schizophrenia [2,4,11,12,13]. Among the available literature, there exists methodological heterogeneity (e.g., study design, evaluation instruments, sample size, age of the participants, inclusion of first episode psychosis, and absence of a control group), as well as equivocal results. For example, Palmer et al. [4] found lower levels of happiness in patients diagnosed with schizophrenia compared to healthy controls, while Agid et al. [11] observed no difference in the level of happiness between these two groups. Another methodological concern is that previous work investigating happiness and related factors in patients with schizophrenia did not include a control group [12,14]. The lack of consensus as to what constitutes happiness as a construct is another source of concern. For example, past research has focused on psychological well-being [2], a concept similar to happiness but more linked to health status [15], with findings demonstrating lower levels of psychological well-being in schizophrenia patients compared with healthy controls.

The relationship between psychiatric symptoms and happiness is also understudied. Symptoms may be associated with happiness in individuals with schizophrenia, but few studies have investigated this association to date [2,4,11,12,13]. One recent study found that negative and depressive symptoms (rather than positive symptoms) were significantly associated with lower levels of happiness [16].

Both happiness and psychological well-being are correlated with overall patient functioning (as well as community and occupational functioning), but this relationship is complex and is shown to be influenced by several variables such as general and social cognition [17,18], higher perceived stress [14], and lower levels of resilience [4]. Interestingly, there is some research that suggests schizophrenia patients with poor functioning have higher levels of happiness and satisfaction with life compared to individuals without a psychiatric diagnosis [13].

Recent studies suggest that the key factors associated with lower levels of psychological well-being in patients with schizophrenia are depressive symptoms, motivational deficits [2,11,13,14], and cognitive disorganization [14]. A recent review found that deficits in well-being in schizophrenia are present prior to the onset of the first episode of psychosis and hypothesized that lower well-being is a risk factor for both the onset of psychosis and poorer functional outcomes [19]. When assessed using a validated objective measure, psychological well-being was positively and significantly related to the strength of the therapeutic alliance between psychiatrists and patients with schizophrenia [20]. Impairments in neurocognition and social cognition have also been extensively studied in patients with schizophrenia [21,22], with results suggesting that cognitive performance is one of the strongest predictors of functioning [18]. To our knowledge, the potential impact of cognition (neurocognition and social cognition) on patients’ happiness has not yet been studied, and this domain was therefore included in the present study.

Although the psychological well-being of patients with schizophrenia has received sufficient attention, subjective happiness has received limited attention in the scientific literature. It seems plausible that patients with schizophrenia have lower levels of happiness than the general population, but a more rigorous analysis of the factors related to happiness in schizophrenia is needed. The primary hypothesis of the present study is that deficits in functioning, social cognition, and neurocognition will predict lower levels of subjective happiness, well-being, and satisfaction with life in patients with schizophrenia. The main objective of the present study is to investigate whether these factors demonstrate stronger relationships than clinical symptoms with outcomes of interest (i.e., subjective happiness, well-being, and satisfaction with life).

## 2. Materials and Methods

This quantitative study utilized a cross-sectional case-control design.

### 2.1. Sample

Overall, 69patients diagnosed with schizophrenia and 87 healthy controls matched on sex, age, and education level participated in the study. In patients, clinical variables (e.g., age of illness onset and duration of illness) and sociodemographic variables (e.g., sex, age, and educational level) were obtained from available medical records. The data collection took place between January and December of 2019.

In the control group, sociodemographic characteristics (e.g., age, sex, and education level) were collected through personal interviews. All data collection took place at the Hospital Universitario Príncipe de Asturias (Alcalá, Madrid, Spain) and at the Hospital San Cecilio (Granada, Spain).

### 2.2. Inclusion and Exclusion Criteria of Participants

Schizophrenia patients were aged 18 to 60, had no psychiatric diagnosis other than schizophrenia according to DSM-5 criteria [23] (including no current diagnosis of substance or alcohol use disorder, excluding caffeine or nicotine), and had no severe, uncontrolled, or unstable medical conditions. Schizophrenia patients were required to have been diagnosed with schizophrenia for at least five years and all participants were engaged in consistent outpatient care at a mental health clinic. Patients were invited to participate in the study at their regular outpatient appointment. After signing the informed consent form and answering all the questions and doubts they wished to ask, the study variables were collected and all the scales included in the procedure were given to them. Schizophrenia patients were not excluded based on psychotropic medication or therapy regimen (i.e., patients included in the study were prescribed a range of psychotropic medications and were engaged in a variety of therapeutic interventions).

Participants in the healthy control group were aged 18 to 60, did not meet DSM-5 criteria for an Axis-I diagnosis according to a clinical interview conducted by a clinical psychiatrist, and were not taking any psychotropic medications. Healthy controls were recruited from the same hospital settings as the schizophrenia patients and were non-treatment-seeking individuals accompanying hospital patients presenting for a variety of treatments (e.g., an individual visiting a family member during recovery from a surgery).

In accordance with the Declaration of Helsinki (1991), all participants in the study signed an informed consent form. Participants did not receive any compensation for participating in this research. This study protocol was approved by the Clinical Research Ethics Committee of the Hospital Universitario Príncipe de Asturias.

### 2.3. Assessment Instruments and Procedure

#### 2.3.1. Variables

Dependent variables: subjective happiness, level of well-being, and satisfaction with life.

Independent variables: sex, age, clinical symptoms (i.e., psychotic and affective), level of insight related to illness, perceived stress, functioning, cognitive performance, and social cognition.

#### 2.3.2. Assessment Instruments

The Subjective Happiness Scale (SHS) [24] is a global self-report measure of happiness. The SHS consists of four items that are averaged for a total score. Two items ask respondents to characterize themselves using absolute and relative intervals. (i.e., on a scale from less happy to very happy), while the other two items offer brief descriptions of happy and unhappy individuals and ask respondents to what extent they identify with each description (i.e., not at all to a great deal). The four items are rated on a Likert scale from 1 to 7. Some examples of items are “Compared to most of my peers, I consider myself” or “Some people are generally very happy. They enjoy life regardless of what is going on, getting the most out of everything. To what to extend does this characterization describe you?” Higher scores reflect higher levels of subjective happiness. A Spanish version of the SHS [25] was administered with adequate reliability observed (α = 0.77).

The Psychological Well-being Scale (SPWB) [26], adapted and validated to Spanish by Díaz and collaborators [27], was used in this study. The SPWB includes six scales derived from 39 items. Participants respond to each item with scores ranging from 1 (strongly disagree) to 6 (strongly agree). Items are summed to create six subscales: self-acceptance, positive relationships with others, autonomy, mastery of environment, purpose in life scale, and personal growth. Higher scores reflect a higher level of self-reported well-being. Some examples of items are: “In general, I feel I am in charge of the situation in which I live” or “I tend to worry about what other people think of me.” All SPWB scales exhibited good internal reliabilities, with Cronbach alpha’s ranging from 0.68 (Personal Growth) to 0.83 (Self-Acceptance).

The Life Satisfaction Scale (SWLS) [28] is a scale consisting of five items measuring self-reported satisfaction with life with demonstrated good internal consistency (Cronbach alpha’s ranging from 0.79 to 0.89). Values of the responses ranged from 1 to 5 according to a traditional Likert Scale where 1 indicates “totally disagree” and 5 indicates “totally agree”. Higher scores reflect higher levels of satisfaction with life. In the Spanish version used [29,30], the reliability analysis showed good internal consistency. Some examples of items are: “In most ways, my life is close to my ideal”, “The conditions of my life are excellent”, or “I am satisfied with my life.”

The Perceived Stress Scale (PSS) [31], with the Spanish adaptation validated by Remor [32], was used in this study and showed adequate reliability (internal consistency, α = 0.81, test-retest, *r* = 0.73), validity (concurrent), and sensitivity. This scale consists of 14 items assessing thoughts and feelings experienced during the past month. Participants endorsed scores between 0 (never) and 4 (very often). Higher scores reflect more perceived stress experienced during the past month.

The Screen for Cognitive Impairment in Psychiatry (SCIP-S) [33] is a tool developed to quantify the nature and severity of cognitive impairment in mental illness. The SCIP-S consists of five tests that assess the following cognitive areas: audio-verbal learning, working memory, verbal fluency, delayed recall, and processing speed. Higher scores reflect more intact cognitive status. Each subscale contains a cut-off point indicating impairment. Alterations in three or more subtests at this cut-off point indicate cognitive impairment. The SCIP-S was used in the present study to analyze the correlations between cognition, happiness, and well-being. This scale was previously used in patients with schizophrenia [34] and the Spanish version has been validated [35]. Test-retest validity was measured with the intraclass correlation coefficient (ICC; values 0.77–0.91).

The Penn Emotion Recognition Task (ER-40) [36] assesses emotion recognition, a domain of social cognition. The ER-40 includes 40 color photographs of faces that express four basic emotions (joy, sadness, anger, and fear) and neutral expressions. There are eight photographs of each expression (four with high intensity and four with low intensity). Participants are asked to correctly identify the emotion expressed in each photograph from five response options. Higher scores indicate better emotion recognition. This scale has been used specifically in patients with schizophrenia and was not translated to Spanish because this task is based mainly on images. The ER-40 was included to examine the relationships between social cognition and levels of life satisfaction and well-being.

Clinical symptomatology was measured by the Positive and Negative Syndrome Scale (PANSS) [37] for the assessment of psychotic symptoms validated in Spanish [38], the Hamilton Depression Rating Scale (HDRS) [39] for the assessment of depressive symptoms validated in Spanish [40], and the Young Mania Rating Scale (YMRS) [41] for the assessment of manic symptoms validated in Spanish [42]. All measures of clinical symptomatology were completed by trained raters.

Functioning was measured by the Functioning Assessment Short Test (FAST) [43], which is designed for the clinical assessment of functional impairment with an excellent internal consistency (Cronbach’s alpha = 0.91). The FAST consists of 24 items grouped into 6 areas of functioning: autonomy, work functioning, cognitive functioning, finance, interpersonal relations, and leisure. Higher scores indicate lower levels of functioning. This scale has been used in patients with schizophrenia with demonstrated internal consistency [44].

Insight was measured through the Scale to Assess Unawareness in Mental Disorder (SUMD) [45]; the SUMD assesses disease awareness in patients with schizophrenia. It provides three scores: overall disease awareness, symptom awareness, and symptom attribution. Higher scores reflect less insight. This scale was administered by clinicians and was validated in Spanish [46]. The Intraclass Correlation Coefficient (ICC) values were all greater than 0.70.

#### 2.3.3. Procedure

All study procedures, including screening for exclusion criteria, were completed in one study session with a duration of approximately 60 min.

### 2.4. Statistical Analysis of Data

Statistical analyses were carried out using SPSS software (version 21). Clinical and sociodemographic variables, as well as independent and dependent variables, such as subjective happiness levels, were compared between patients with schizophrenia and healthy controls using Student’s *t*-tests (continuous variables) or Chi-square tests for categorical variables (e.g., sex). A series of bivariate correlations (Pearson’s *r*) were conducted among variables of interest prior to regression analyses, including subjective happiness, total psychological well-being, and perceived stress. The degree of statistical significance for all hypothesis-contrast tests was set at *p* < 0.05.

A series of multiple linear regression models with dependent variables of subjective happiness (SHS), level of well-being (SWLS), and satisfaction with life (SPWB) were conducted using the stepwise backward technique. This technique allowed us to enter all independent variables exhibiting significant correlations with dependent variables of interest and interpret the most parsimonious models. The aim of this sets of analyses was to investigate relationships between clinical symptomatology (measured with the PANSS, HDRS, YMRS, and SMUD scales), global functioning (using the FAST scale), perceived stress (measured with the PSS), cognitive function (measured with the SCIP), and social cognition (ER-40 scale) with the three dependent variables while controlling for confounding variables. The parameters used to estimate the strength of the associations with dependent variables were the coefficient of partial correlation (partial *r*) and the coefficient of determination (adjusted *R*^2^).

## 3. Results

### 3.1. Sociodemographic and Clinical Characteristics

The patient group was composed of 69 patients diagnosed with schizophrenia according to DSM-5 criteria and was predominantly male (47 of 69 (68.1%)). The mean age (±SD) of schizophrenia patients was 41.8 years (±11.4).

The control group was composed of 87 participants without a psychiatric diagnosis according to DSM-5 criteria. The mean age of control participants was 44.9 years (±13.3) with slightly more females than males (50 of 87 (57.5%)).

When comparing groups, a higher percentage of females, married people, and more years of education were observed in the control group with worse physical health indices observed in the patient group. Table 1 summarizes sociodemographic and clinical characteristics.

Schizophrenia patients in the present study had an average illness duration of 18 years with a history of recurrent psychotic episodes (6.5 on average) and inpatient hospitalizations (4.7 on average). The majority of patients also reported past suicidal ideation (56.5%), and there was a relatively high rate of substance use (e.g., 53–68%) compared with the control group, a good level of adherence to treatment (81.2%), and a higher proportion of patients receiving psychotherapy (65.2%). Results of the clinical scales that were administered (i.e., PANSS, SUMD, YMRS, and HDRS) are summarized in Table 2.

### 3.2. Correlations between Happiness and Clinical Variables

Bivariate correlations examining relationships among clinical variables in the patient group demonstrated significant positive correlations between happiness, life satisfaction, and well-being, and a significant negative correlation between perceived stress and subjective happiness, life satisfaction, and functioning. Perceived stress and age demonstrated a significant negative correlation, suggesting that younger patients had higher levels of perceived stress. Finally, a significant positive correlation between cognitive function (measured with the SCIP scale) and social cognition (measured with the ER-40 scale) was observed. All correlations and significance values are presented in Table 3.

When we analyzed the relationship between SHS (subjective happiness) and FAST scores (difficulties in daily life) in patients, we observed that there was an inverse and significant correlation (*r* = −0.44; *p* = 0.01) in the subgroup of patients without cognitive impairment (i.e., less than three subscales of the SCIP battery impaired); when the same analysis was performed in patients with cognitive impairment (i.e., three or more subscales of the SCIP battery impaired), this correlation was not significant (*r* = −0.18; *p* = 0.30).

### 3.3. Comparisons between Groups

When comparing groups across clinical variables, we found that the control group had significantly higher levels of subjective happiness (measured with the SHS scale), life satisfaction (measured with the SWLS scale), and psychological well-being (measured with the SPWB scale) compared with schizophrenia patients. In addition, patients had a significantly higher level of perceived stress (measured with the PSS scale) and significantly greater difficulty accurately recognizing emotions (measured with the ER-40 scale) compared with healthy controls. See Table 4 for all group comparisons with significance tests.

With regards to cognition, schizophrenia patients demonstrated significantly lower performance on the SCIP tool compared with the control group indicating more impairment (see Table 4). Schizophrenia patients exhibited significantly lower SCIP total scores as well as significantly lower scores on the subscales of immediate and delayed verbal learning, verbal fluency, and processing speed. Schizophrenia patients also had lower scores on the working memory subscale but this difference was not statistically significant.

### 3.4. Variables Associated with Happiness and Well-Being in Schizophrenia Patients

Results of reduced multiple linear regression models are presented in Table 5. In the model with subjective happiness as a dependent variable, a significant partial correlation was observed with the SWLS total score (standardized regression coefficient β = 0.35) and the SPWB total score (β = −0.61), as well as the SPWB subscales of self-acceptance (β = 0.51), positive relationship with others (β = 0.35), and environmental mastery (β = 0.37). There was a negative partial correlation observed between subjective happiness and the FAST total score (β = −0.27). This model explained 37% of the variance, suggesting good predictability within behavioral sciences.

When we analyzed the factors associated with well-being (measured by SPWB scale) we found a significant relationship with subjective happiness (β = 0.38; adjusted R^2^ = 0.14). In the model with life satisfaction as the dependent variable, significant partial correlations were observed with predictors of self-acceptance (β = 0.39), purpose in the life (β = 0.21; measured by SPWB scale), and domains of cognition including immediate verbal learning (β = 0.35) and processing speed (β = 0.25). SPWB autonomy (β = −0.27), perceived stress (β = −0.25), and delayed verbal learning (β = −0.44) were also significant predictors of life satisfaction with observed negative correlations. This model explained more than 59% of variance in life satisfaction (see Table 5).

Interestingly, scores on clinical variables (i.e., PANSS, HDRS, and YMRS), physical health variables (BMI and hours of sleep), and insight level (measured by SMUD scale) did not demonstrate significant relationships with the dependent variables of interest (i.e., subjective happiness, level of well-being, and satisfaction with life).

## 4. Discussion

### 4.1. Main Findings

As expected, schizophrenia patients exhibited lower levels of subjective happiness (measured by the SHS scale), life satisfaction (SWLS scale), and psychological well-being (SPWP scale) compared with healthy controls. Schizophrenia patients also demonstrated more impaired cognition (SCIP scale) and social cognition (ER-40) as well as higher levels of perceived stress (PSS scale) compared with the control group. In addition, patients with schizophrenia exhibited lower levels of functioning compared with healthy controls.

A series of multivariate linear regressions allowed us to investigate the role of several potential predictors of subjective happiness, life satisfaction, and well-being in patients with schizophrenia. In line with our main hypothesis, we found that better functioning significantly predicted higher levels of subjective happiness. Additionally, in line with our main hypothesis, we found that lower levels of impaired cognition significantly predicted higher levels of life satisfaction. Our hypothesis was not supported when examining well-being, since neither functioning nor cognitive variables were significant predictors of this outcome. In addition to cognition and functioning, results from this study also highlight the significant role of perceived stress predicting satisfaction with life and the interplay between subjective happiness, life satisfaction, and well-being. Altogether, results from this study underscore the impact of domains of cognition and functioning on levels of subjective happiness and life satisfaction in patients with schizophrenia.

Since the main scales used (SHS, SWLS, and SPWB) do not have cut-off points, we were not being able to perform logistic regression models that would allow us to determine the influence of certain qualitative variables (sociodemographic factors, substance use, hours of sleep, physical exercise, or adherence to pharmacological treatment) on the main dependent variables analyzed.

### 4.2. Clinical Implications

Schizophrenia is a complex disease, characterized by severe impairment in many areas of daily life, including the ability to maintain social relationships, hold a job, and live independently. Our results are consistent with those of Fervaha and colleagues [2] that people with schizophrenia maintain a good outlook on life despite high rates of functional impairment. Schizophrenia patients in the present study demonstrated high scores on measures of subjective happiness, satisfaction, and well-being, despite relatively high rates of functional impairment.

However, despite high scores on measures of happiness, satisfaction, and well-being, schizophrenia patients still demonstrated lower average levels of happiness when compared with scores of healthy controls. These results are consistent with those of other studies [2,4], but in contrast with Agid and colleagues [11], who found that schizophrenia patients did not endorse different levels of happiness compared with healthy controls. The discrepancy between these findings could be due to the influence of other variables on the patients’ perception of happiness. In our study, higher levels of happiness in the control group were associated with higher levels of life satisfaction and sense of well-being (Table 5).

When analyzing the association between subjective happiness and functioning, we found that this relationship was influenced by cognitive impairment. In schizophrenia patients without cognitive impairment, subjective happiness and level of functioning were significantly positively correlated. However, in schizophrenia patients with cognitive impairment, there was no significant relationship observed between subjective happiness and level of functioning. This result is consistent with Izydorczyk and colleagues [14], who found an association between psychological well-being and cognitive disorganization.

This finding has important clinical implications and suggests that consideration of patient cognitive functioning is an important factor when treatment planning. Targeting subjective happiness or functioning, or both of these domains, in schizophrenia patients while considering level of cognitive impairment may result in improved treatment outcomes [17]. Additionally, targeting cognitive impairment may be a first step towards improving subjective happiness. To our knowledge, this is the first study investigating the relationship between cognitive impairment and subjective happiness. One important clinical implication of this finding is better understanding of a subgroup of patients with cognitive impairment in whom low levels of happiness do not impact overall functioning. Given that the study of happiness and life satisfaction in patients with severe mental illness has received little attention, results from this study highlight the role of cognition as a potential treatment target to increase well-being in schizophrenia.

Findings from the current study also highlight the role of perceived stress in subjective happiness and well-being. Patients with schizophrenia exhibited higher levels of perceived stress. Perceived stress was found to be inversely related to life satisfaction, even after controlling for other variables, suggesting that perceived stress may be another promising treatment target to achieve improved recovery and treatment outcomes.

In contrast with other studies [2,11,13,14,16], we did not find an association between subjective happiness, psychological well-being, satisfaction with life, and psychiatric symptoms (e.g., depressive and negative symptoms). Potential reasons for this discrepancy may be that individuals in the current study exhibited a wide range of psychiatric symptoms (versus including only patients who are euthymic or in a state of symptom remission), patients with schizophrenia in the current study had a longer duration of illness compared with previous studies (e.g., compared with studies focused on first-episode psychosis), as well as sample size considerations (e.g., current study included more participants than some previous studies), or use of multivariate analyses, which simultaneously consider psychiatric symptoms alongside other clinical predictors (e.g., perceived stress and functioning).

### 4.3. Strengths and Limitations

Our study has both strengths and limitations to consider. One main strength of the present study is the inclusion of schizophrenia patients and healthy controls with a relatively large sample sizes; to our knowledge, this is one of the largest studies investigating predictors of happiness in schizophrenia. Two additional strengths of the present study are the inclusion of schizophrenia patients with psychiatric symptoms, which improves the generalizability of results, and the use of multivariate analyses to simultaneously consider multiple variables and present parsimonious models highlighting the most salient predictors of happiness, well-being, and life satisfaction.

The major limitation of this study is its retrospective and cross-sectional design, which precludes conclusions regarding directionality of the relationships observed, as well as the use of clinical interviews instead of structured interviews to diagnose the control group. Additionally, course-of-illness and age-of-onset data were obtained retrospectively. Although we attempted to confirm reports, when possible, with clinical records, treating physicians, and family members, some recall bias may have influenced the results. Another limitation of the present study is heterogenous therapy and psychotropic medication regimens among schizophrenia patients. For ethical reasons, it was not possible to ask patients to refrain from receiving psychopharmacological and/or psychotherapeutic treatment and these treatment effects may have influenced our results.

### 4.4. Future Lines of Research

Future studies with a prospective longitudinal design are needed to investigate directionality of the relationships among subjective happiness, well-being, life satisfaction, and other domains of interest from this study such as functioning, cognitive impairment, and social cognition. Additionally, future studies should seek to recruit larger sample sizes to apply analyses that may better elucidate the complex relationships among predictors of interest from this study (e.g., interactions between functioning, cognitive impairment, and perceived stress). To this end, multicenter studies may identify additional sociodemographic variables (e.g., socioeconomic status and country of origin) that impact levels of subjective happiness in individuals with schizophrenia. Other important factors, such as family support and overload caregivers [47,48], have not been analyzed in our research and should be included in the future.

Although it is true that psychopharmacological and psychosocial treatments are increasingly sophisticated and broadly focused on the full recovery model to improve quality of life in individuals with schizophrenia, the results of this study (i.e., schizophrenia patients showed lower levels of subjective happiness compared with healthy controls) and the relatively nascent area investigating subjective happiness in schizophrenia suggest that this is an important area warranting future research and clinical attention. Achieving functional improvement and improving levels of subjective happiness and life satisfaction is an important treatment target to consider, regardless of the presence or absence of current psychotic symptoms.

## 5. Conclusions

In conclusion, patients with schizophrenia endorsed lower levels of subjective happiness, well-being, and life satisfaction compared with healthy controls. Cognitive impairment was a significant predictor of lower satisfaction with life. Additionally, significant positive relationships existed between functioning and subjective happiness in schizophrenia patients without cognitive impairment. However, this relationship was not present in patients with cognitive impairment, suggesting cognition may modulate the relationship between functioning and subjective happiness. Higher levels of functioning and cognitive impairment and lower levels of perceived stress were also shown to be significant predictors of subjective domains of happiness and well-being in schizophrenia. While subjective happiness and other related outcomes may be endorsed at lower levels in schizophrenia patients, this study identified several potential treatment targets (e.g., functioning, perceived stress, and cognition) to increase subjective happiness and enhance overall recovery in schizophrenia.

## Figures and Tables

**Table 1 ijerph-18-07706-t001:** Description of the differences between the healthy control group and schizophrenia group according to sociodemographic and clinical variables.

	Healthy Control Group (*n* = 87)	Schizophrenia Patients (*n* = 69)	Test Statistic	*p-*Value
Mean age (SD)	44.9 (13.3)	41.8 (11.4)	−1.47 ^1^	0.143
Sex (*n* (%))				
Male	37 (42.5)	47 (68.1)	10.14 ^2^	0.001
Female	50 (57.5)	22 (31.9)		
Marital status (*n* (%))				
Currently married	45 (51.7)	3 (4.3)	40.55 ^2^	<0.001
Currently unmarried	42 (48.3)	66 (95.7)		
Years of education (SD)	13.7 (3.0)	10.9 (2.2)	−6.64	<0.001
Work status (*n* (%))				
Disabled (temporarily/permanent)	2 (2.3)	24 (34.8)	83.24 ^2^	<0.001
Unemployed	8 (9.2)	34 (49.3)		
Working (full/part-time)	77 (88.5)	11 (15.9)		
BMI (SD)	24.4 (3.1)	26.7 (4.7)	3.48 ^1^	0.001
Living arrangement (*n* (%))				
Family of origin or on their own	68 (78.2)	66 (95.7)	84.54 ^2^	<0.001
Family created by the participant	19 (21.8)	3 (4.3)		
Physical comorbidity (*n* (%))				
Yes	4 (4.6)	20 (29.0)	17.58 ^2^	<0.001
Tobacco consumption (*n* (%))				
Yes	29 (33.3)	49 (71.0)	21.85 ^2^	<0.001
Coffee consumption (*n* (%))				
Yes	59 (67.8)	48 (69.6)	0.05 ^2^	0.815
Alcohol consumption (*n* (%))				
Regular use	15 (17.2)	37 (53.4)	22.92 ^2^	<0.001
Illegal drugs consumption (*n* (%))				
Yes	5 (5.7)	47 (68.1)	67.36 ^2^	<0.001
Active days per week (SD)	4.3 (2.2)	4.2 (2.2)	−0.257 ^1^	0.797
Hours of sleep (*n* (%))				
<8 h per night	81 (93.1)	38 (55.1)	30.76 ^2^	<0.001
8 or more hours per night	6 (6.9)	31 (44.9)		
Intensity of exercise (*n* (%))				
Low	23 (26.4)	22 (31.9)	2.47 ^2^	0.291
Moderate	51 (58.4)	32 (46.4)		
Intense	13 (14.9)	15 (21.7)		

BMI: body mass index; SD: standard deviation. ^1^ Student’s *t*-test, ^2^ Chi-square test.

**Table 2 ijerph-18-07706-t002:** Description of clinical variables in the schizophrenia patient group.

Clinical Variables and Tests Cores	Schizophrenia Patients (*n* = 69)
Duration of illness (years), Mean (SD)	18.0 (9.5)
Age of illness onset, Mean (SD)	24.0 (7.0)
Age of diagnosis, Mean (SD)	25.8 (7.4)
Age of initial hospital admission, Mean (SD)	27.5 (8.9)
Numbers of psychotic episodes, Mean (SD)	6.5 (5.1)
Number of hospital admissions, Mean (SD)	4.7 (4.6)
PANSS Positive score, Mean (SD)	12.8 (5.3)
PANSS Negative score, Mean (SD)	14.4 (6.2)
PANSS General score, Mean (SD)	28.6 (10.6)
PANSS Total score, Mean (SD)	55.7 (20.2)
HDRS Total score, Mean (SD)	7.3 (4.8)
YMRS Total score, Mean (SD)	3.6 (4.4)
SUMD Total score, Mean (SD)	21.4 (15.9)
FAST Total score, Mean (SD)	22.9 (13.9)
Suicide ideation in the past (*n* (%))	
Yes	39 (56.5)
No	30 (43.5)
Suicide attempts in the past (*n* (%))	
Yes	26 (37.7)
No	43 (62.3)
Number of suicide attempts, Mean (SD)	1.4 (1.3)
Substance abuse (*n* (%))	
No abuse	22 (31.9)
Abuse in the past	11 (15.9)
Current abuse	36 (52.2)
Adherence to medication treatment (*n* (%))	
Good	56 (81.2)
Moderate	10 (14.5)
Poor	3 (4.3)
Received psychotherapy (*n* (%))	
Yes	45 (65.2)
Never	24 (34.8)

SD: standard deviation. PANSS: Positive and Negative Syndrome Scale; HDRS: Hamilton Depression Rating Scale; YMRS: Young Mania Rating Scale; SUMD: Scale to Assess Unawareness in Mental Disorder; FAST: Functional Assessment Staging Test.

**Table 3 ijerph-18-07706-t003:** Correlations between clinical variables in schizophrenia patients.

Tests Scores	Schizophrenia Patients (*n* = 69)					
	SHS Score	SWLS Score	SPWB Score	PSS Score	SCIP Score	ER-40 Score
SHS score		0.528 ***	0.362 ***	−0.272 *	0.023	−0.050
SWLS score			0.268 *	−0.425 ***	−0.015	−0.081
SPWB score				−0.144	−0.036	−0.071
PSS score					0.192	0.059
SCIP score						0.472 ***
FAST score	−0.315 *	−0.115	−0.206	−0.251 *	−0.068	0.026
Age	0.042	0.197	0.034	−0.432 **	−0.304 *	−0.078
BMI	0.150	0.075	0.108	−0.072	−0.228	−0.161
PANSS score	−0.150	−0.112	−0.012	0.045	−0.306	−0.234

* *p* < 0.05; ** *p* < 0.005; *** *p* < −001. SHS: Subjective Happiness Scale; SWLS: Satisfaction with Life Scale; SPWB: Scale of Psychological Well-Being; PSS: Perceived Stress Scale; SCIP: Screen for Cognitive Impairment in Psychiatry; FAST: Functional Assessment Staging Test; ER-40: Emotion Recognition Task; BMI: body mass index; PANSS: Positive and Negative Syndrome Scale.

**Table 4 ijerph-18-07706-t004:** Description of the differences between the healthy control and schizophrenia group according to happiness, well-being, functioning, and cognition outcomes.

Tests Scores	Healthy Control Group (*n* = 87)	Schizophrenia Patients (*n* = 69)	Test Statistic	*p-*Value
SHS Total score, Mean (SD)	4.9 (0.8)	4.2 (1.3)	−3.54 ^1^	<0.001
SWLS Total score, Mean (SD)	18.9 (3.7)	14.4 (5.6)	−5.77 ^1^	<0.001
SPWB Total score, Mean (SD)	153.4 (16.0)	145.3 (22.2)	−2.56 ^1^	0.012
SPWB Self-acceptance	23.4 (4.2)	22.2 (4.6)	−1.71 ^1^	0.088
SPWB Positive relationship with others	23.8 (5.2)	21.3 (5.7)	−2.93 ^1^	0.004
SPWB Autonomy	28.0 (5.5)	28.6 (5.7)	0.635 ^1^	0.526
SPWN Environmental Mastery	24.5 (3.5)	23.5 (8.1)	−1.01 ^1^	0.317
SPWB Personal Growth	27.0 (5.5)	27.3 (4.6)	0.39 ^1^	0.698
SPWB Purpose in life	26.8 (4.2)	23.6 (5.8)	−3.83 ^1^	<0.001
PSS Total score, Mean (SD)	20.3 (7.4)	26.2 (7.8)	4.75 ^1^	<0.001
SCIP Total score, Mean (SD)	84.8 (13.9)	68.4 (16.6)	−6.71 ^1^	<0.001
SCIP Verbal learning test immediate	22.7 (3.2)	17.8 (4.7)	−7.52 ^1^	<0.001
SCIP Working memory	19.9 (3.5)	17.6 (5.7)	−2.89 ^1^	0.005
SCIP Verbal Fluency	22.9 (6.5)	18.6 (5.5)	−4.44 ^1^	<0.001
SCIP Verbal learning test delayed	6.8 (2.0)	4.5 (2.7)	−5.86 ^1^	<0.001
SCIP Processing speed	12.6 (4.0)	9.5 (4.2)	−4.76 ^1^	<0.001
Cognitive impairment (*n* (%))				
No (impairment in ≤2 SCIP scales)	73 (83.9)	32 (46.4)	24.63 ^2^	<0.001
Yes (impairment in ≥3 SCIP scales)	14 (16.1)	37 (53.6)		
ER-40 Total score, Mean (SD)	31.9 (3.6)	30.3 (4.3)	−2.42 ^1^	0.017

SHS: Subjective Happiness Scale; SWLS: Satisfaction with Life Scale; SPWB: Scale of Psychological Well-Being; PSS: Perceived Stress Scale; SCIP: Screen for Cognitive Impairment in Psychiatry; ER-40: Emotion Recognition Task; SD: standard deviation. ^1^ Student’s *t* test, ^2^ Chi-square test.

**Table 5 ijerph-18-07706-t005:** Multiple linear regressions of factors associated with happiness and well-being in 69 schizophrenia patients.

Variables	Standardized Regression Coefficient (β)	t_exp_	*p* Value
Subjective Happiness Scale (SHS) *			
SWLS	0.346	2.820	0.007
SPWB Total score	−0.613	−2.027	0.047
Self-acceptance	0.515	2.474	0.016
Positive relationship with others	0.352	2.148	0.036
Environmental mastery	0.368	2.158	0.035
FAST Total score	−0.275	2.229	0.030
Scale of Psychological Well-Being (SPWB) **			
SHS	0.380	3.178	0.002
SCIP Working memory	−0.228	−1.905	0.062
Scale of Satisfaction with Life (SWLS) ***			
SPWB			
Self-acceptance	0.386	3.583	0.001
Autonomy	−0.274	−2.963	0.005
Purpose in life	0.214	2.015	0.049
PSS Total Score	−0.253	−2.658	0.010
SCIP Verbal learning test immediately	0.348	2.929	0.005
SCIP Verbal learning test delayed	−0.436	−3.145	0.003
SCIP Processing speed	0.248	2.082	0.042

* Coefficient of determination (adjusted R^2^) = 0.366, F = 0.924, df = 1, 54, *p* = 0.341. ** Coefficient of determination (adjusted R^2^) = 0.144, F = 1.795, df = 1, 58, *p* = 0.186. *** Coefficient of determination (adjusted R^2^) = 0.598, F = 2.282, df = 1, 53, *p* = 0.137. *Note:* SHS: Subjective Happiness Scale; SWLS: Satisfaction with Life Scale; SPWB: Scale of Psychological Well-Being; PSS: Perceived Stress Scale; SCIP: Screen for Cognitive Impairment in Psychiatry; FAST: Functional Assessment Staging Test; df: degrees of freedom.

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
