# Peer review of "Functioning and Happiness in People with Schizophrenia: Analyzing the Role of Cognitive Impairment"

_ijerph, 2021, doi:10.3390/ijerph18147706_

Round 1
Reviewer 1 Report
This is a very interesting attitude to analyze a correlation between cognitive functioning in schizophrenia, however the manuscript needs several improvements:
- In the manuscript there are several English language mistakes, which need correction. For example:
"All schizophrenia patients were between the ages of 18 and 60"
- The following exclusion criterion is not clear enough for me. Can you make it more precise?
"Schizophrenia patients were not excluded based on psychotropic medica- 114 tion or therapy regimen"
- Can you clarify this criterion?
"Healthy controls were recruited from 120 the same hospital settings as the schizophrenia patients and were typically individuals 121 accompanying hospital patients presenting for a variety of treatments"
- I do not understand following sentences, could you clarify? Do you mean "accept, approve", etc.
"Two additional strengths are the inclusion of schizophrenia patients endorsing psychiatric symptoms..."
"(schizophrenia patients endorse lower levels of subjective happiness compared with healthy controls)"
- In the discussion of factors influencing happiness in schizophrenia it is also important to mention family support, e.g.:
Şahin, F. and Şahin Altun, Ö., 2020. The relationship between perceived family support and happiness level of patients with schizophrenia. Journal of Psychiatric Nursing, 11(3), pp.181-187.
Author Response
Reviewer 1
This is a very interesting attitude to analyze a correlation between cognitive functioning in schizophrenia, however the manuscript needs several improvements:
1. In the manuscript there are several English language mistakes, which need correction. For example: "All schizophrenia patients were between the ages of 18 and 60"
The authors, including a co-author speaking English as a first language, have reviewed the manuscript, and made edits throughout (highlighted in yellow) to improve presentation of the results in English.
2. The following exclusion criterion is not clear enough for me. Can you make it more precise? "Schizophrenia patients were not excluded based on psychotropic medication or therapy regimen"
More precise language has been added to this exclusion criteria (Material and Methods section):
“Schizophrenia patients were not excluded based on psychotropic medication or therapy regimen (i.e., patients included in the study were prescribed a range of psychotropic medications and were engaged in a variety of therapeutic interventions).”
3. Can you clarify this criterion? "Healthy controls were recruited from the same hospital settings as the schizophrenia patients and were typically individuals accompanying hospital patients presenting for a variety of treatments"
More precise language has been added to this description to clarify (Material and Methods section):
“Healthy controls were recruited from the same hospital settings as the schizophrenia patients and were non-treatment-seeking individuals accompanying hospital patients presenting for a variety of treatments (e.g., an individual visiting a family member during recovery from a surgery).”
4. I do not understand following sentences, could you clarify? Do you mean "accept, approve", etc.
"Two additional strengths are the inclusion of schizophrenia patients endorsing psychiatric symptoms..."
More precise language has been added to this sentence for clarification (in the Discussion section):
“Two additional strengths of the present study are the inclusion of schizophrenia patients with psychiatric symptoms, which improves the generalizability of results, and the use of multivariate analyses to simultaneously consider multiple variables and present parsimonious models highlighting the most salient predictors of happiness, wellbeing, and life satisfaction.”
"(schizophrenia patients endorse lower levels of subjective happiness compared with healthy controls)"
Language has been edited for same tense to improve clarity:
“schizophrenia patients showed lower levels of subjective happiness compared with healthy controls”
5. In the discussion of factors influencing happiness in schizophrenia it is also important to mention family support, e.g.:
Thank you very much for your comment. We have included in the Discussion section the next sentence, as well as two new references:
“Other important factor, as family support and overload caregivers [47, 48] have not been analyzed in our research and should be included in the future”.
Two new references:
- Quah, S. Caring for persons with schizophrenia at home: examining the link between family caregivers’ role distress and quality of life. Sociol Health Illn. 2014; 36, 596-612. https://doi.org/1111/1467-9566.12177.
- Şahin, F., Şahin Altun, Ö. The relationship between perceived family support and happiness level of patients with schizophrenia. J Psy Nurs. 2020; 11, 181-187. https://doi.org/10.14744/phd.2020.09821
Reviewer 2 Report
I appreciate the opportunity to review this manuscript. Overall, I find the manuscript to investigate an important syndrome. The manuscript is also well-presented and well-written. The project is appropriately designed and executed. I only have trivial comments and questions to the authors.
Could the authors explain why the study chooses to focus on cognitive impairment and not other psychotic symptoms associated with schizophrenia with regards to functioning and happiness?
When did the research take place? Could you assume some external events could contribute to the response, such as COVID-19?
Author Response
Reviewer 2
I appreciate the opportunity to review this manuscript. Overall, I find the manuscript to investigate an important syndrome. The manuscript is also well-presented and well-written. The project is appropriately designed and executed. I only have trivial comments and questions to the authors.
Thank you very much for your interest and for your nice comments.
Could the authors explain why the study chooses to focus on cognitive impairment and not other psychotic symptoms associated with schizophrenia with regards to functioning and happiness?
The principal reason to choose the cognitive impairment is that this factor had received less attention to know the relationship between functioning and happiness.
We have included it in the Introduction section:
“Impairments in neurocognition and social cognition have also been extensively studied in patients with schizophrenia [21,22], with results suggesting that cognitive performance is one of the strongest predictors of functioning [18]. To our knowledge, the potential impact of cognition (neurocognition and social cognition) on patients' happiness has not yet been studied, and this domain was therefore included in the present study.”
And in the Discussion section:
“To our knowledge, this is the first study investigating the relationship between cognitive impairment and subjective happiness. One important clinical implication of this finding is better understanding of a subgroup of patients with cognitive impairment in whom low levels of happiness do not impact overall functioning. Given that the study of happiness and life satisfaction in patients with severe mental illness has received little attention, results from this study highlight the role of cognition as a potential treatment target to increase well-being in schizophrenia.”
When did the research take place? Could you assume some external events could contribute to the response, such as COVID-19?
Our research take place between January and December of 2019, we have included this date in the Methods section. In that sense COVID-19 pandemic have not any influence in our data collection. The influence of stress is similar to any other study with these patients prior to the pandemic.
Reviewer 3 Report
In their paper, " Functioning and happiness in people with schizophrenia: Analyzing the role of cognitive impairment", The authors report on data from quantitative observations utilizing a cross-sectional case-control study that had as objective to identify whether these factors may have more influence than the presence of clinical symptomatology in the dependent variables studied. The results suggest that rehabilitation programs may improve recovery outcomes with a focus on subjective happiness and functioning, especially in patients with cognitive impairment.
The findings of this study are of relevance for health promotion, particularly as potential information for the development of effective actions in this patient group.
In my opinion, the article is well written except for a few minor spell checks required.
I have particular comments or suggestions.
Methods
1. For the instruments used, please add items’ examples
2. The procedure is very brief and does not detail what was done in the study.
3. Procedures for statistical analyses are well described and clear for the readers.
Discussion
It is of interest, the topic is appealing and the quality of writing and discussion is good.
Author Response
Reviewer 3
In their paper, " Functioning and happiness in people with schizophrenia: Analyzing the role of cognitive impairment", The authors report on data from quantitative observations utilizing a cross-sectional case-control study that had as objective to identify whether these factors may have more influence than the presence of clinical symptomatology in the dependent variables studied. The results suggest that rehabilitation programs may improve recovery outcomes with a focus on subjective happiness and functioning, especially in patients with cognitive impairment.
The findings of this study are of relevance for health promotion, particularly as potential information for the development of effective actions in this patient group.
Thank you very much for your interest and nice comments.
In my opinion, the article is well written except for a few minor spell checks required.
The authors have revised the manuscript to correct spelling mistakes and to improve presentation of the results in English. Edits to improve clarity of writing are highlighted in yellow throughout the manuscript.
I have particular comments or suggestions.
Methods
- For the instruments used, please add items’ examples
We have included some examples in the principal scales used. Now you can read in the Methods section:
Subjective Happiness Scale (SHS) [24]: is a global self-report measure of happiness. The SHS consists of four items that are averaged for a total score. Two items ask respondents to characterize themselves using absolute and relative intervals. (i.e., on a scale from less happy to very happy), while the other two items offer brief descriptions of happy and unhappy individuals and ask respondents to what extent they identify with each description (i.e., not at all to a great deal). The four items are rated on a Likert scale from 1 to 7. Some examples of items are: “Compared to most of my peers, I consider myself” or “Some people are generally very happy. They enjoy life regardless of what is going on, getting the most out of everything. To what to extend does this characterization describe you?” Higher scores reflect higher levels of subjective happiness. A Spanish version of the SHS [25] was administered with adequate reliability observed (α = .77).
Psychological Well-being Scale (SPWB) [26], adapted and validated to Spanish by Díaz and collaborators [27]: The SPWB includes six scales derived from 39 items. Participants respond to each item with scores ranging from 1 (strongly disagree) to 6 (strongly agree). Items are summed to create six subscales: self-acceptance, positive relationships with others, autonomy, mastery of environment, purpose in life scale, and personal growth. Higher scores reflect a higher level of self-reported well-being. Some examples of items are: “In general, I feel I am in charge of the situation in which I live” or “I tend to worry about what other people think of me.”All SPWB scales exhibited good internal reliabilities, with Cronbach alpha’s ranging from 0.68 (Personal Growth) to 0.83 (Self-Acceptance).
Life Satisfaction Scale (SWLS) [28]: a scale consisting of five items measuring self-reported satisfaction with life with demonstrated good internal consistency (Cronbach alpha’s ranging from 0.79 to 0.89). Values of the responses ranged from 1 to 5 according to a traditional Likert Scale where 1 indicates "totally disagree" and 5 indicates "totally agree". Higher scores reflect higher levels of satisfaction with life. In the Spanish version used [29, 30], the reliability analysis showed good internal consistency. Some examples of items are: “In most ways, my life is close to my ideal”, “The conditions of my life are excellent” or “I am satisfied with my life.”
- The procedure is very brief and does not detail what was done in the study.
Now we have included more details in the Methods section (all the changes are highlighted in yellow):
Schizophrenia patients were aged 18 to 60, had no psychiatric diagnosis other than schizophrenia according to DSM-5 criteria [23] (including no current diagnosis of substance or alcohol use disorder, excluding caffeine or nicotine), and had no severe, uncontrolled or unstable medical conditions. Schizophrenia patients were required to have been diagnosed with schizophrenia for at least five years and all participants were engaged in consistent outpatient care at a mental health clinic. Patients were invited to participate in the study at their regular outpatient appointment. After signing the informed consent form and answering all the questions and doubts they wished to ask, the study variables were collected and all the scales included in the procedure were given to them. Schizophrenia patients were not excluded based on psychotropic medication or therapy regimen (i.e., patients included in the study were prescribed a range of psychotropic medications and were engaged in a variety of therapeutic interventions).
Participants in the healthy control group were aged 18 to 60, did not meet DSM-5 criteria for an Axis-I diagnosis according to a clinical interview conducted by a clinical psychiatrist, and were not taking any psychotropic medications. Healthy controls were recruited from the same hospital settings as the schizophrenia patients and were non-treatment-seeking individuals accompanying hospital patients presenting for a variety of treatments (e.g., an individual visiting a family member during recovery from a surgery).
- Procedures for statistical analyses are well described and clear for the readers.
Thanks for the comment
- Discussion
It is of interest, the topic is appealing and the quality of writing and discussion is good.
Thank you very much for the appreciation.
Round 2
Reviewer 1 Report
I would like to thank the Authors for all changes and improvements they have done. I have no further comments.